# Physics-Aware Downsampling with Deep Learning for Scalable Flood Modeling

**Niv Giladi**[1,2]   **Zvika Ben-Haim**[1]   **Sella Nevo**[1]   **Yossi Matias**[1]   **Daniel Soudry**[2]

[1]Google Research
[2]Technion - Israel Institute of Technology

```
{giladiniv, daniel.soudry}@gmail.com
{zvika, sellanevo, yossi}@google.com
```

## Abstract

**Background.** Floods are the most common natural disaster in the world, affecting the lives of hundreds of millions. Flood forecasting is therefore a vitally important endeavor, typically achieved using physical water flow simulations, which rely on accurate terrain elevation maps. However, such simulations, based on solving partial differential equations, are computationally prohibitive on a large scale. This scalability issue is commonly alleviated using a coarse grid representation of the elevation map, though this representation may distort crucial terrain details, leading to significant inaccuracies in the simulation.

**Contributions.** We train a deep neural network to perform physics-informed downsampling of the terrain map: we optimize the coarse grid representation of the terrain maps, so that the flood prediction will match the fine grid solution. For the learning process to succeed, we configure a dataset specifically for this task. We demonstrate that with this method, it is possible to achieve a significant reduction in computational cost, while maintaining an accurate solution. A reference implementation accompanies the paper as well as documentation and code for dataset reproduction.

## 1   Introduction

A physical process is a time-evolving phenomenon marked by gradual transitions through a series of states, following physical rules. Scientists attempt to model these physical processes by employing analytical descriptions that correspond to the underlying physics of these processes. Commonly, the analytical descriptions, or physical models, such as conservation laws, are in the form of partial differential equations (PDEs). Physical models are the backbone behind modeling phenomena such as weather forecasts, flood forecasting, aerodynamics, chemical processes, and more. Traditionally, these physical models are solved numerically, as no closed form solution is available for most problems. One typical numerical method is to discretize the PDEs using a Taylor expansion and repeatedly integrate in time over short intervals. However, solving PDEs numerically on a large scale is challenging because of the need to resolve spatiotemporal features over many scales.

In this paper, we focus on hydraulic models, which are a widely used physical models. The goal of these models is to simulate the flow of water in a given environment. We are specifically interested in simulating water flow through a floodplain in order to predict inundation, i.e., where and to what extent will flooding occur. Specifically, in inundation modeling we are given an elevation map of the terrain, boundary conditions, initial conditions — and we wish to predict, the water height in each pixel at some future time $t$, with maximum accuracy and minimum computational cost. Such prediction is an important task in operational flood alert systems [5, 43, 10].

Unfortunately, inundation modeling is often computationally intractable when modeling flow on a large scale with fine grid resolution. A coarse grid representation of the elevation map can significantly improve computation time, for two reasons. First, less computation is done at each step of the PDE solver. Specifically, in a discretized PDE the per-step spatial computation is done between neighboring pixels. Therefore, if we downsample a $d$-dimensional map by a factor of $x$, we decrease this per-step computation by a factor of $x^d$. Second, the maximum stable time step in the numerical solution can be increased by a factor of $x$, as entailed by the Courant–Friedrichs–Lewy (CFL) condition [23, 7]. Taking both factors into account, we gain a total theoretical speedup of $x^{d+1}$. For example, in this paper we will show that for the two-dimensional problem ($d = 2$) of inundation modeling, we retain fairly good accuracy when downsampling by a factor of 16, leading to a significant $4096\times$ speedup.

So far it was considered challenging to coarse-grain a general elevation map without losing accuracy in the inundation prediction. In particular, small details in the elevation map can significantly impact the water flow evolution. Narrow canals, dams, and embankments (barriers preventing a river from flooding an area) are examples of such details that require special care. There are several typical methods to represent an elevation map on a coarse grid, all of which can cause a sub-optimal representation, as small, important details might get lost [11, 44, 43]. For example, when applying standard downsampling methods, an 8 meter wide embankment might be reduced to half its original height above its neighboring surroundings in a 16 meters resolution coarse map. More complex downsampling techniques can be employed, but in practice, to achieve high-quality simulations at a coarse resolution, domain expertise is required, along with a deep understating of the underlying physics of the specific problem.

To address these limitations, we optimize the coarse grid representation of the elevation map so that its water height solution will match the accurate fine grid solution. This is accomplished by combining a deep neural network (DNN) with the PDEs describing the flow of water. The DNN downsamples the fine grid elevation map, and the downsampled elevation map is fed into the PDEs, where each time step is a recurrent unit, to calculate the water height solution. Thus, the DNN is optimized directly on the water height solution, through the PDEs. This also requires back-propagating through the PDEs, whose time-steps need to be unrolled. We demonstrate that this approach can yield a better coarse grid representation than conventional methods, in terms of water flow estimation. To this end, we configured a dataset designed specifically for inundation modeling. Each sample consists of a fine grid elevation map, boundary conditions, and a simulation time. The ground truth of each sample is the corresponding fluid state, calculated on a fine grid. This dataset is available along with code for further use.

The contributions of our work include:

- A new framework for learning how to coarse-grain a PDE together with its external environment — by minimizing the distance between the solutions on the fine and coarse grids (Sec. 4.1).

- A novel configured dataset for 2D water flow estimation, with accessible code for reproduction[1]. (Sec. 4.2)

- A demonstration that gradients backpropgated for *tens of thousands* of steps through the PDE remain informative (Sec. 5.1) and can pinpoint key features (Fig. 2)

- A demonstration of the accuracy of our approach in inundation modeling, for two different application settings (Sec. 5.2 and 5.3). Since we decrease the map resolution by a factor 16, this entails a $4096\times$ speedup. Such a speedup has a significant impact on the feasibility of real-time flood warning for large populations (Sec. 8).

## 2   Related Work

The integration of machine learning with physical models, described by PDEs, has emerged in recent years as a useful tool for efficiently solving computational science problems. In particular, the use of machine learning models is promising in problems where little is known about the underlying process, or in problems where the computational overhead is significant [42].

**Data driven models.** Sharifi et al. [32], Vandal et al. [40], Gentine et al. [14], and Srivastava et al. [36] use basic ML methods to interpolate between coarse and fine representations to improve accuracy

---

[1]The code for this paper is available at https://github.com/tech-submissions/physics-aware-downsampling

in modeling. In these studies, they use supervised data driven models that predict the solution of the physical model. Data driven models can fail to capture important complex relationships between physical variables directly from data, and it is necessary to ensure that the learned models follow physical laws.

**Physics constrained loss function.** To force the predictive models to follow physical laws, a common technique is to incorporate the physical constraints as additional terms in the loss function [19, 29, 30, 35, 4]. These physics-informed DNNs have several advantages. First, the computation of the physical loss term can be done without labeled data. In addition, the physical loss term acts as a regularizer, so it can reduce the search space of the optimization. These models has shown success in several tasks such as forward solving of PDEs and inverse problems [42]. However, physical loss terms require fine-tuning to find the optimal weighting between the predictive loss and the physical loss.

**Physics guided architecture.** Another common methodology is physics guided architecture, where domain knowledge is utilized to design architectures that capture physics dependencies among variables [33, 12, 9, 17, 24]. These models are more interpretable than regular models. However, the architecture design can be useful for a specific type of PDEs, and make no sense for others. Designing a specific architecture for each physical model can be a tedious task of its own.

**Data driven discretization.** Perhaps the most closely related works showed that modeling features of the fluid dynamics and numerical scheme with neural networks can enable solving PDEs on much coarser grids than is possible with standard numerical methods [4, 15, 21, 38]. By learning to modify the PDE solver, these works enable using a coarser grid (potentially with some overhead) while retaining accuracy and numerical stability. However, they did not examine how to optimally coarse grain the external environment (given as input to the solver) — as we do here, for terrain elevation maps in the context of flood modeling. In this case, it is critical that the coarse-grained map retains the relevant features, to prevent drastic differences between fine and coarse solutions (e.g., floods). In other words, these previous works address the discretization error entailed in numerical modeling, while this paper suggests a method to reduce the input error, such as errors in the elevation map. The two approaches (modifying the PDE solver vs. modifying the external environment) are orthogonal, and can be potentially combined in future work. Empirically, we found that the input error is dominant in our setting of flood modeling, however, it is an interesting question to quantify this more precisely. Note our approach has no computational overhead during inference since we use the same solver, just on a coarser grid with a modified map. Evidently, we provide experiments on a significantly larger scale than these previous works.

## 3 Physical Model: The Shallow Water Equations

In this section we describe the physical model of fluid flow used in this paper, i.e. the hydraulic model. At the most fundamental level, fluid flow is described by the Navier–Stokes equations. In practice, solving these equations directly is often infeasible, and simpler variations are derived for specific problem settings. One such variation is the shallow water equations [22], a set of partial differential equations that describe water flow when water depth is much smaller than horizontal extent, as is generally the case for floods. The shallow water equations are derived by depth-integrating the Navier–Stokes equations [41, 8, 7]. These assumptions allow a considerable simplification in the mathematical formulation and numerical solution, which enables analysis on much larger horizontal length scales.

To write the shallow water equations, we first define the following notation: $t$ is the time index, $(x, y)$ are the spatial horizontal coordinates, $h(x, y; t)$ is the water height relative to the terrain elevation $z(x, y)$, $\mathbf{q} = (q_x(x, y; t), q_y(x, y; t))$ is the horizontal mass flux of the fluid, $\nabla = (\frac{\partial}{\partial x}, \frac{\partial}{\partial y})$ is the horizontal gradient operator, $n$ is Manning's friction coefficient, and $g$ is the gravitational acceleration constant. The shallow water equations are described by two conservation laws. First, the conservation of mass, which gives rise to the continuity equation,

$$\frac{\partial h}{\partial t} + \nabla \cdot \mathbf{q} = 0\,. \tag{1}$$

Second, the conservation of momentum, which yields the momentum equation,

$$\frac{\partial \mathbf{q}}{\partial t} + gh\nabla(h + z) + \frac{gn^2\,\|\mathbf{q}\|}{h^{7/3}}\mathbf{q} = 0. \tag{2}$$

Note that eqs. 1 and 2 neglect advection and the Coriolis force, as is common for flood modeling. We follow De Almeida & Bates [7] to numerically approximate eqs. 1 and 2 from the continuous domain into the discrete domain, where the numerical solution uses a finite difference scheme applied to a staggered grid. In the discretized domain, functions becomes arrays (e.g., $z(x,y)$ becomes $\mathbf{z}$).

In real-world flood forecasting settings, data for the elevation map would be collected ahead of time, and the boundary conditions for the model would be provided either by real-time measurements or hydrologic forecasts. One would then run the hydraulic model (eqs. 1 and 2) in real-time, or offline on a wide range of possible boundary conditions. The simulation provides both the flood extent and the inundation depth at every pixel. This information can then be used to inform both authorities working on mitigation and response efforts, as well as individuals that may be affected by the expected flooding (e.g., alert apps on mobile phones). However, to do this, the simulation must be done on a large scale and fast enough to be practically relevant. Next, we show how this can be done using physics-aware downsampling of the elevation map.

# 4   Learning to Downsample the Elevation Map

In this section, we formulate the map downsampling problem for several application settings and define the training setup. Then, we describe data generation and configuration. Lastly, we describe the architecture considerations and optimization settings which enable the learning process success.

## 4.1   Problem Setup

Consider a downsampling DNN $f_W$ and a numerical water flow solver $S$ with fixed target time $t$. The downsampling network is a deep neural network parameterized by weights $W$. It transforms a two dimensional fine grid elevation map $\mathbf{z} \in \mathbb{R}^{m \times m}$ into a coarse grid elevation map $\hat{\mathbf{z}} \in \mathbb{R}^{k \times k}$ via the mapping $\hat{\mathbf{z}} = f_W(\mathbf{z})$. The fine and coarse grid water heights, $\mathbf{h} \in \mathbb{R}^{m \times m}$ and $\hat{\mathbf{h}} \in \mathbb{R}^{k \times k}$ respectively, are computed by the numerical solver $S$ with initial and boundary conditions $\mathbf{c}$ such that

$$\mathbf{h} = S(\mathbf{z}, \mathbf{c}) \,, \; \hat{\mathbf{h}} = S(\hat{\mathbf{z}}, \mathbf{c}) \,.$$

A loss function $\ell(\mathbf{h}, \hat{\mathbf{h}})$ penalizes any difference between the fine grid water height $\mathbf{h}$ and the coarse grid water height $\hat{\mathbf{h}}$. The fine grid water height is downsampled to a coarse grid for calculation of the loss. Note that this differs from calculating the water height on a coarse grid, since the water height is calculated on a fine grid and only then is downsampled to a coarse grid. Since we are interested in inundation modeling in coarse resolution, the deviations in coarsening the fine water height are acceptable and can be accomplished by simple averaging of the fine water height. Figure 1 depicts the system overview. Our first general goal is to find a mapping $f_W$ through the minimization problem:

$$W^* = \arg\min_W \mathbf{E}_{\mathbf{z}, \mathbf{c}} \left[ \ell \left( S(\mathbf{z}, \mathbf{c}), S(f_W(\mathbf{z}), \mathbf{c}) \right] \,. \tag{3}$$

The second goal we consider is a weaker variation of eq. (3), which is relevant in operational flood forecasting systems, as described in Sec. 5.2. In such a setting, the fine grid elevation map is known, and we wish to find a coarse representation of the elevation map that will generalize to many boundary conditions. We condition the expectation on $\mathbf{z}$ and the minimization problem can be rewritten as:

$$W^*(\mathbf{z}) = \arg\min_W \mathbf{E}_{\mathbf{c}} \left[ \ell \left( S(\mathbf{z}, \mathbf{c}), S(f_W(\mathbf{z}), \mathbf{c}) \mid \mathbf{z} \right] \,. \tag{4}$$

To train and evaluate a model based on the minimization problem in eq. 3, training and validation data is required, which we next discuss how to configure.

## 4.2   Dataset configuration

**Elevation map curation.** In recent years, improvements in remote sensing and data processing at large scales have improved the access to reliable, diverse, and rich earth-science data. A case in point is the extensive collection of 1-meter Digital Elevation Models (DEMs) published by the United States Geological Survey (USGS) [1]. These DEMs are created using airborne lidar in 1-meter × 1-meter cell size resolution, and cover most of the United States, including flood-prone areas around the Mississippi River and the Arkansas River [13, 27]. We based our dataset on terrain data from the floodplains of those rivers in three different locations. These areas are a mere fraction of the available

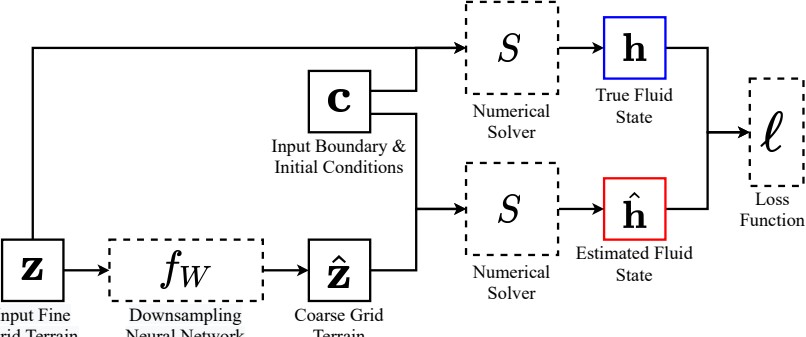

Figure 1: **Training setup for the downsampling neural network** $f_W$**.** Each data sample consists of a fine grid elevation map $\mathbf{z}$, with boundary and initial conditions $\mathbf{c}$. Two solutions are calculated - the fine grid water height $\mathbf{h}$ and the coarse grid water height $\hat{\mathbf{h}}$. The latter uses a coarse grid version of the input elevation map $\hat{\mathbf{z}}$. We train the downsampling neural network $f_W$, to minimize the loss measuring the distance between the two solutions (eqs. 3 or 4). This is done by backpropagating the loss gradients through the numerical solver.

USGS data, but we focused on them due to compute resource limitations. The data was further processed into tiled elevation maps. Mountainous terrains and areas with missing data were filtered to be appropriate for the hydraulic simulation. Each elevation map contains 2000×2000 pixels, covering a 2km×2km region. We found this size is large enough to capture the way differences between coarse and fine grid solutions can affect flooding even in distant locations. We gathered a total of 5183 elevation maps. For each elevation map, we defined boundary conditions as influx, outflux and discharge. More details about data configuration can be found in Sec. A.1 of the appendix.

**Ground truth calculation.** For the scope of this paper, we wish to calculate the flow of the water, i.e., the water height at each pixel, over each data sample, given its boundary conditions over some time $t$. It is possible to construct the corresponding ground truth by simulating the water flow over the fine grid elevation maps. For each elevation map, its ground truth is a 2000×2000 matrix, describing the water height at each pixel. This is a computationally demanding task, but it can be performed offline once. We provide access to the full dataset along with code to reproduce the data and expand to more data samples.

### 4.3 Architecture

We use the ResNet-18 architecture [16] as the core of our DNN model. The last layer of the classifier was removed, and a $1 \times 1$ convolution was used instead to transform the last hidden layer with multiple channels, into a single channel output. In our experiments, we use a downsampling factor of $16\times$, which is natively implemented in the ResNet-18 model. For a smaller downsampling factor it is possible to remove all layers with resolution lower than needed and apply the $1 \times 1$ convolution for a single channel output. It is also possible that other models can perform as well, but these kinds of experiments are beyond the scope of this paper.

For the output scheme, we first notice that simple average coarsening techniques works reasonably well for many of the elevation maps and for a large portion of the spatial domain. This is because fine important features effecting the hydraulic flow are relatively scarce. Therefore, we suggest that the downsampling DNN $f_W$ output will consists a summation of an average pooling coarse grid elevation map and a correction map given by the ResNet-18 model. This is implemented by adding an average pooling skip connection from the input of the ResNet-18, directly to its output. This has two main advantages. First, the ResNet-18 output values have a smaller range and variance, in comparison to the full coarse grid elevation maps. Second, the starting point of the DNN accuracy is close to the accuracy of average pooling downsampling. Both advantages help the learning process converge significantly faster and make performance less sensitive to initialization — in comparison to the case where the DNN outputs the entire coarse grid elevation map rather than a correction (details in the Appendix).

**Data normalization.** It is common practice to normalize the input data of the DNN, e.g., by performing standardization, min-max normalization, and more. However, using multiplicative normalization such as standardization will distort the vertical scale of the elevation map and essentially remove important information. Indeed, two elevation maps can have the same normalized form, but encompass different hydraulic properties, and possibly different optimized coarse grid representation. Consequently, we normalize the input elevation map only by subtracting its mean.

### 4.4 Optimization scheme

The per-sample loss is a sum of the per-pixel loss $\ell$ on all the map pixels $\sum_{i,j} \ell(h_{i,j}, \hat{h}_{i,j})$. Here $\ell$ is Huber loss[2] which exhibited good convergence behavior. The training loss was optimized using Adam [20], and a batch size of 32 samples.

There are several challenges in training the DNN in the setting described in Sec. 4.1. First, to obtain gradients for each sample, one must calculate the hydraulic solution, i.e., iteratively applying the PDEs in the forward pass, and then calculate the gradients in the backward pass, through the PDEs. The backward calculation through the PDEs takes the majority of the iteration time. Ideally, we would like to reach a steady state solution, where the velocity field and total mass do not change. However, to keep the training computationally feasible, we stop the solver after it finished simulating one hour of hydraulic flow. This still might require thousands of recurrent iterations of the PDEs, depending on the CFL condition described in Sec. 3.

**Gradient checkpointing.** As discussed in Sec. 4.1, we define each fine grid elevation map to be relatively large, $2000 \times 2000$ pixels, to allow long-distance flow effects to manifest. As a result, the activation maps are large, causing a memory bottleneck and limiting the batch size that can be used in each node. In addition, the flow of water calculation might take thousands of recurrent iterations, as described in Sec. 1, increasing both memory and compute time. To address this computational challenge, we use the gradient checkpointing mechanism, as described in Siskind & Pearlmutter [34] and Chen et al. [6]. The checkpointing is applied both to the DNN, and to the numerical solver.

**Numerical stability of the PDEs.** In addition to instabilities originating in DNN non-convex optimization, one needs to take into account numerical instability of the PDEs when performing a learning rate scan. The numerical scheme of the Shallow Water Equations is prone to numerical instability issues, much like other numerical schemes. This instability is more apparent when there are large variations in the terrain, causing rapid changes in the water velocity and depth. When training with learning rates larger than $10^{-2}$ with Adam, and $10^{-1}$ with Stochastic Gradient Descent (SGD), divergence problems occur, especially at the beginning of training: over-correction by the gradients cause topographic exaggeration by the model which leads to numerical instability of the PDEs. To avoid this, we performed our learning rate scan with in the relatively low range of $[10^{-3}, 10^{-1}]$ to find the empirically optimal learning rate in terms of convergence rate and generalization, as well as numerical stability of the PDEs. Adam achieved better generalization and therefore is used.

## 5 Experiments

In this section, we leverage our physics-aware DNN to accurately solve the shallow water equations on a coarse grid, over a variety of elevation maps and boundary conditions. Specifically, we:

1. Demonstrate that the system (Sec. 4.1) can locate hydraulically significant details of the elevation map, using the accumulated gradient after backpropagation through the PDEs.

2. Optimize the DNN over a single elevation map with multiple boundary conditions (eq. 4) and demonstrate better performance compared to traditional downsampling. This is a typical setting in operational flood modeling systems, where repeated simulations are performed with different boundary conditions.

3. Train the DNN over a training set of different elevation maps and boundary conditions (eq. 3) and show generalization capabilities on both elevation maps and boundary conditions. This would enable fast downsamping of maps in new areas.

---

[2]i.e. $\ell(h, \hat{h}) = 0.5(h - \hat{h})^2$ if $|h - \hat{h}| \leq 1$ and $\ell(h, \hat{h}) = \left| h - \hat{h} \right| - 0.5$ otherwise.

Throughout the experiments, we use an averaging baseline for comparison. This baseline is intuitive for terrain elevation map downsampling, and is being used in traditional inundation modeling studies [11, 43]. We also consider slightly less naive downsampling techniques, and we demonstrate that they are not superior to the baseline we use (Appendix A.2.3). More details on the experiments can be found in Appendix A.2.

## 5.1 Backpropagation through PDEs

In this section, we demonstrate that hydraulically meaningful details in the elevation map are identified by backpropagation through the PDEs, allowing the model to learn to preserve these details when downsampling. Specifically, we show that the gradient signal can properly flow even through *tens of thousands* of unrolled recurrent units representing the PDE time-steps. We explicitly construct an example to demonstrate that important details in the elevation map can be detected by the gradients: We start with an elevation map with an evident embankment, and artificially flatten a few pixels of that embankment. The modified elevation map is shown in Fig. 2a with a red square around the flattened pixels (a zoomed-in version of the red square is shown in Fig. 2b-top). From the boundary conditions we used, water flows in from the right bottom part of the elevation map and flows out from right upper part. With the flattened pixels, water can flow through the embankment, inundating a large area left of the embankment which is not inundated in the non-modified elevation map. This difference is shown in Fig. 2d, where red pixels are more inundated on the modified elevation map solution (over-flooding), and blue pixels are more inundated on the non-modified elevation map solution (under-flooding). In turn, the loss is high because of the large inundated area. A meaningful gradient will identify the discrepancy as resulting from the flattened embankment pixels, rather than the entire inundated region. The resulting gradient map of the elevation map after backpropagation is shown in Fig. 2c. As can be seen, the gradient is highest at the flattened pixels, especially compared to the inundated region, indicating that those few pixels contributed significantly to the deviation from the non-modified elevation map solution. We emphasize that it was required to unroll more than ten thousand iterations to compute the gradient, as we simulated 6 hours of hydraulic flow.

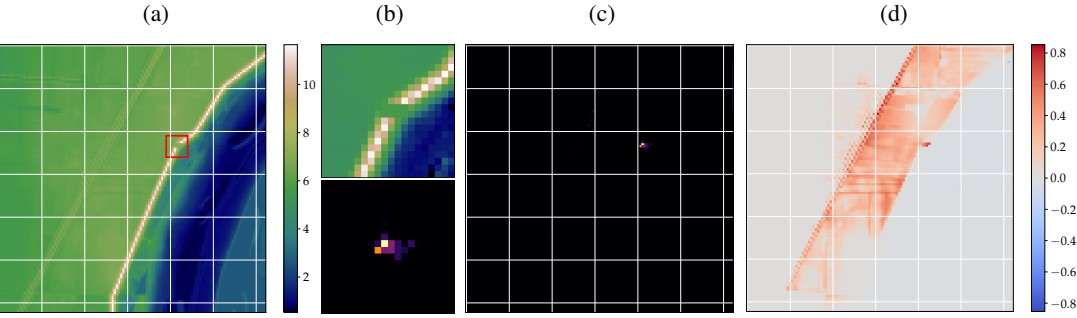

Figure 2: **Gradient flow through PDEs.** (**a**) A modified elevation map with a few pixels of the embankment flattened . (**d**) Difference map between the solution calculated on the modified elevation map and the non-modified elevation map solution. (**c**) Gradient magnitude of the modified elevation map, computed by back-propagating through the PDEs. (**b**) Zoom-in on the modified pixels of the elevation map (top) and on the gradient magnitude map of the same location (bottom).

## 5.2 Generalization over multiple boundary conditions

In this section, we find an optimized coarse grid representation of a single elevation map with varying boundary conditions, as in the minimization problem in Eq. (4). To do so, we optimize a DNN where each data sample uses the same elevation map, but with different boundary conditions — including influx and outflux locations as well as discharge. This setting is useful for repeated simulations with different boundary conditions over the same spatial domain, a typical setting in operational flood modeling systems [5]. We validate the model on randomly sampled boundary conditions. The baseline we compare ourselves against remains average pooling. In Figure 3 we show a comparison between the water flow solution on a fine grid elevation map and each of the solutions calculated on coarse grid elevation maps, both for the trained DNN and baseline. As can be seen, solutions calculated on the DNN coarse grid elevation map are closer to the fine grid solution. We provide more examples on different elevation maps in Appendix A.2.4.

In Figure 4 we provide example of visible features preserved by the DNN. On the right image, there is a small embankment that is preserved entirely in the DNN version, while in the baseline version the embankment is averaged with its surroundings, causing it to almost disappear. On the left image, there is a narrow canal with embankments on either side. Again, the DNN preserves both the canal and most of the embankments, while in the baseline version, the canal is averaged with the embankments, causing partial flattening of the terrain at the canal location. We emphasize that the DNN coarse grid elevation maps are outputs of two different models that were trained on each image separately.

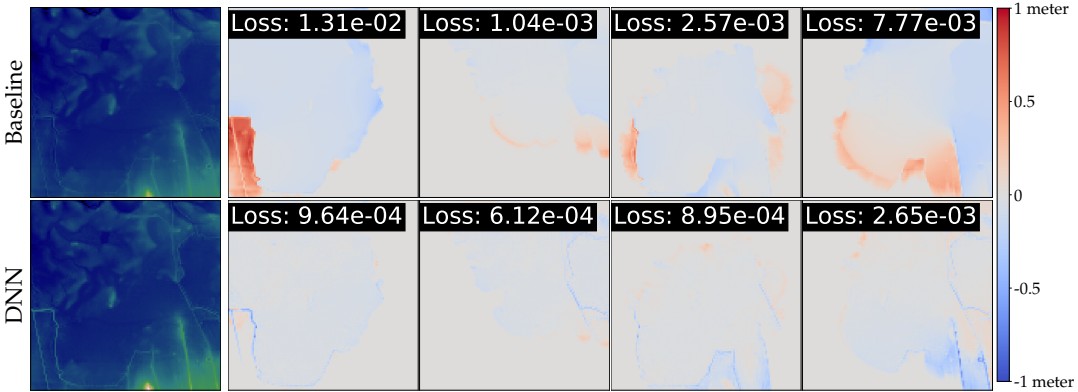

Figure 3: **Generalization over different boundary conditions.** Leftmost images are coarse grid elevation maps downsampled using a DNN (bottom) and average pooling baseline (top), where the DNN trained on different boundary conditions and the same elevation map. Each column is a comparison of solutions between the DNN and the baseline elevation maps, for different boundary conditions. Each image on those columns is a difference map between the coarse and fine grid solutions along with the Huber loss. Red indicates pixels more inundated in the coarse grid solution, and blue indicates pixels more inundated in the fine grid solution. The color scale, in meters, appears on the right. Solutions calculated on the DNN elevation map achieve better accuracy compared to the baseline.

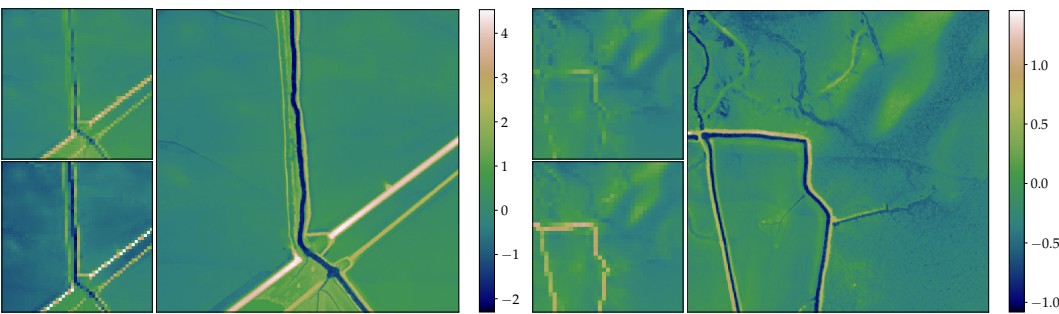

Figure 4: **Capturing fine details important to water flow.** Two patches taken from different elevation maps. The two larger images are the fine grid patches and the two smaller images on the left of each fine grid image are the coarse grid patches of the average pooling baseline (top) and the DNN (bottom). Note the preservation of hydraulically significant features such as canals and embankments in the DNN as opposed to the baseline.

## 5.3 Generalization over different elevation maps

For scalable inundation modeling, a robust model for coarse grid representation is required, which is able to generalize to elevation maps and boundary conditions unseen during training. We train such model on 4031 elevation maps, each with different boundary conditions. We validate the model on 1155 unseen elevation maps, each with different boundary conditions. We compare the accuracy of the water flow solutions obtained on a coarse grid between the DNN elevation maps and the average pooling elevation maps. The metric used for water flow solution accuracy is Huber loss. We used Adam with a learning rate $10^{-3}$, and a batch size of 32 samples. The training was done in 90 epochs. Figure 5a shows the convergence plot of the model and its improvement over the baseline. Note that

the model starts with a loss similar to the baseline, because of the architecture scheme described in section 4.3.

Next, we plot the 2D histogram of the DNN's loss compared to the baseline (Figure 5b). The resulting distribution is almost entirely below the linear dashed line, indicating better accuracy by the DNN, matching Figure 5b. There is a large number of data points with small loss (both by the model and the DNN), meaning that for a large portion of the data, average pooling achieves reasonable accuracy. Overall, the modified elevation maps exhibited an improvement of the Huber loss by a factor of 2 over all the tested elevation maps. This improvement increases to a factor of 2.5 as we look only at harder examples (90% percentile). Other accuracy metrics show similar results with a larger improvement factor of up to 3. Additional analysis is provided in Appendix A.2.5.

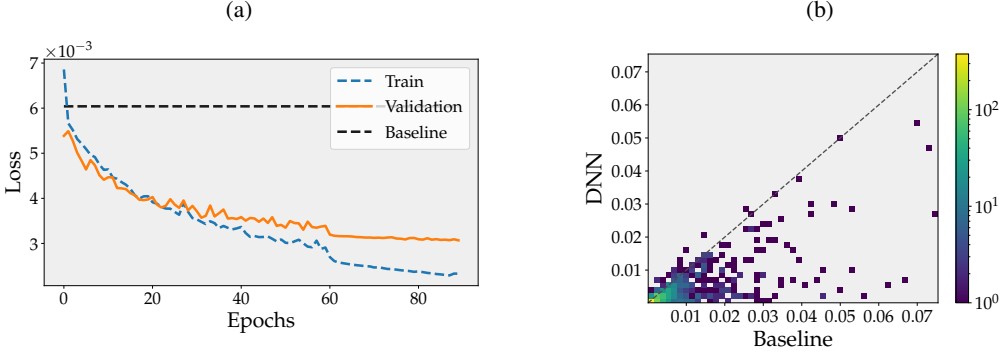

(a)  (b)

Figure 5: **Generalization over different elevation maps.** (**a**) - Convergence plot of the model loss for both training and validation. (**b**) - A comparison of the per-sample Huber loss values of the baseline (x-axis) and the DNN (y-axis). The color scale is logarithmic. The DNN model surpass the accuracy of the baseline, with an improvement of the Huber loss by a factor of 2. Additional analysis is provided in Appendix A.2.5

## 6  Summary

Enabling accurate inundation modeling is a vitally important endeavor for flood forecasting, and physical water flow simulations are typically a core component of that endeavor. These simulations rely on accurate elevation maps, and computationally demanding on a large scale. Elevation map coarsening can significantly reduce the computation required, but hydraulically important fine details might get distorted on a coarse grid. We tackle this problem by training a physics aware DNN that downsamples elevation maps to a coarse grid representation, while preserving details important for hydraulic simulation. We do so by incorporating a physical model into the learning process, such that gradients are calculated through the PDEs. We also configure a dataset specifically for this task and make it available for further use. We consider two applicable training settings for operational flood forecasting systems, and demonstrate that such training is feasible and that the learned DNN outputs meaningful elevation maps with better water flow results compared to prevalent methods.

## 7  Broader impact

Floods cause thousands of fatalities, affect hundreds of millions of people, and cost approximately $10 billion every year [10, 39]. Flood early warning systems have the potential to significantly reduce both fatalities and economic costs, in many cases between 30-50% [31, 25, 2, 28]. Inundation models are a critical component of such systems, as many people will not take protective action without information that is spatially accurate [5]. However, the vast majority of fatalities and harms occur in low and middle income countries, where both financial resources and expertise to support such efforts is scarce [10, 26]. As a result, the requirement of enormous computational resources, or the expertise to intelligently downscale elevation maps to enable more efficient simulation, can be infeasible for the most important uses of these models. This work can bypass the need for either of these constraints, providing a tool for the resource-constrained to use in their modeling, or helping support efforts to scale up global inundation models, such as those pursued by organized by Google, Fathom, ECMWF, CGIAR and others.

## 8 Acknowledgments

We thank Elad Hoffer for technical advising, as well as Yaniv Blumenfeld and Mor Shpigel Nacson for valuable comments on the manuscript. The research of DS was supported by the Israel Science Foundation (grant No. 1308/18), and by the Israel Innovation Authority (the Avatar Consortium).

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
