# A Supplementary Material

## A.1 Data Configuration

The inputs to a hydraulic simulation include an elevation map, initial conditions, and the boundary conditions. For a given elevation map, there is an infinite possible combinations of initial and boundary conditions that could potentially realize in future events. It is an interesting question how to automatically configure the most relevant initial and boundary conditions to train on, to get a representation that will be useful in potential future real-world scenarios. We suggest a basic configuration that adequate for the purpose of this paper.

**Initial conditions.** These include the water height $\mathbf{h} \in \mathbb{R}^{m \times m}$ at each pixel and a staggered grid flux $\mathbf{q} \in \mathbb{R}^{2 \times (m-1) \times (m-1)}$ in each direction $x, y$. For simplicity, since we are interested in the steady state solution so the initial conditions is less relevant, we used zero initial conditions, meaning no water at the beginning of the simulation. We note that it is possible to obtain physically reasonable initial conditions by taking snapshots of $\mathbf{h}$ and $\mathbf{q}$ along a simulation and use it as initial conditions.

**Boundary conditions.** These include the locations and widths where water flows in and out along with the discharge of the water that flows in (influx) and the slope of the water that flows out (outflux). It is not obvious how to automatically define reasonable realistic boundary conditions. For example, for a single river crossing the elevation map, it would make sense to define the influx at the upstream edge of the river and accordingly at the downstream edge of the river. It gets more complicated as river branches appear, and even more complicated where no river exists at all, as happens in many of the elevation maps. We suggest a basic heuristic approach to automatically define the boundary conditions. First, we find the lowest point on the boundary of the elevation map, and define this point as the outflux location. Then, the influx location is defined to be the middle of the opposite axis of the chosen outflux axis. We define it this way to get a large flood extent, by constraining the water to cross the elevation map before it can flow out. We define both the influx and outflux widths to be 400 meters. The data configured with this method is used in Section 5.3. In Section 5.2, a different method is used where we define multiple boundary conditions for the same elevation map, as described in Section A.2.4.

As described in Section 3, we follow De Almeida & Bates [7] to discretize the shallow water equations. The discrete implementation described in De Almeida & Bates [7] uses two parameters – a weighting factor $\theta$ that adjusts the amount of artificial diffusion, and a coefficient $0 < \alpha \leq 1$ that is used as a factor by which we multiply the time step. We use the proposed values in De Almeida & Bates [7], namely $\theta = 0.7, \alpha = 0.7$.

## A.2 Experiments Details

### A.2.1 Technical details

The experiments were conducted on two configurations of machines. For experiments described in Section 5.2, we used distributed data parallelism on a machine with 4 Nvidia RTX2080TI GPUs. For the experiments described in Section 5.3 we used distributed data parallelism on a machine with 8 Nvidia RTX2080TI GPUs.

**Handling a varying number of iterations.** At each iteration $t$, the time step $dt^{(t)}$ depends on the current maximum water height, $\max_{i,j} h_{i,j}^{(t)}$, following the CFL condition described in Section 3. For different boundary conditions and elevation maps, $\max_{i,j} h_{i,j}^{(t)}$ can vary, yielding a different time step for each iteration $t$, and ultimately a different number of iterations until the required time of water flow simulation is achieved. In batch processing, each sample in the batch finishes the forward pass calculation in a different number of iterations. To resolve this issue, we perform iterations with $dt^{(t)} = 0$ for samples that finished their calculation until all samples in the batch are done.

### A.2.2 Choosing the loss function

For the optimization process we examined two loss functions: the mean squared error (MSE)

$$\ell_{\text{MSE}}(h, \hat{h}) = (h - \hat{h})^2$$

and the Huber loss (also referred as smooth $\ell_1$)

$$\ell_{\text{Huber}}(h, \hat{h}) = \begin{cases} 0.5(h - \hat{h})^2 & , \text{if } |h - \hat{h}| \leq 1 \\ \left| h - \hat{h} \right| - 0.5 & , \text{if } |h - \hat{h}| > 1 \end{cases}$$

summing the loss over samples and pixels. In addition, for test purposes we defined a new heuristic loss which aims to capture meaningful floods for inundation modeling purposes (similarly to how the 0-1 loss captures meaningful errors in classification problems). This new loss, called "Inundation Error", equals 1 if the difference between the predicted and ground true water height is above a certain threshold parameter $c$, and 0 otherwise:

$$\ell_{\text{Inundation}}(h, \hat{h}; c) = \begin{cases} 1, & \text{for } |h - \hat{h}| \geq c \\ 0, & \text{otherwise} \end{cases}$$

The motivation for this new loss is based on the fact that, in flood forecasting warning systems, areas inundated above some water height must be evacuated, regardless of the exact water height. Therefore, in this setting, errors larger than some predetermined water height level have the same meaning for inundation modeling. A value $c = 0.5$ meter is used for the inundation error, as a real threshold estimate for flood forecasting systems.

Fig. A.1 shows a comparison between the three loss functions, when using the baseline average pooling. As can be seen, the loss values of all loss functions are highly correlated. Therefore, the training can potentially be done with either of the first two differentiable loss functions. However, the correlation of the inundation error with the Huber loss is somewhat higher than the MSE. Also, the Huber loss exhibited better convergence properties. Therefore, it was used for training.

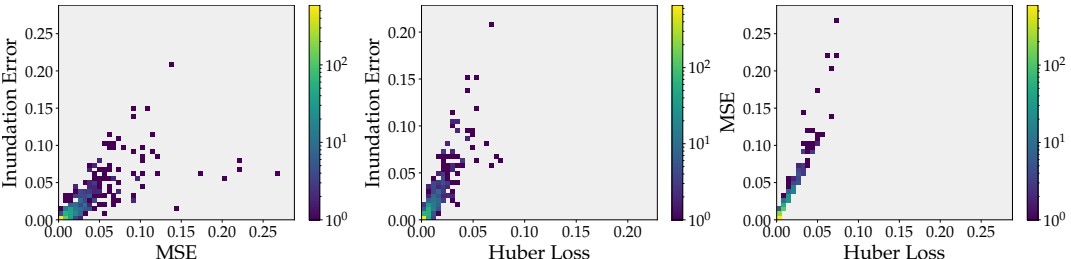

Figure A.1: **Correlation between loss metrics.** A comparison of the per-sample loss values for average pooling with different loss functions. The colorscale is logarithmic. The corresponding Pearson correlations from left to right are 0.78, 0.88, and 0.97

### A.2.3 Edge preserving downsampling

It is reasonable to ask whether there exist downsampling techniques that perform better than average pooling, but do not require the heavy machinery of deep learning. In this experiment, we show two slightly less naive downsampling techniques and demonstrate that they do not successfully capture the intricacies of hydraulically meaningful downsampling. Specifically, we use a bilateral filter [37] to smooth the elevation map while preserving edges, before downsampling. We then downsample the map using either average pooling or max-pooling. The rationale behind max-pooling is that we seek to preserve high-elevation features such as embankments, whose height is reduced by average pooling.

However, as shown in Fig. A.2 and A.3, neither of these techniques outperforms the naive average pooling method used in the main text. This reinforces the claim that machine learning is an appropriate tool for hydraulically-aware downsampling.

### A.2.4 Generalization over multiple boundary conditions

In this section, we provide more details of the experiment depicted in Section 5.2. The DNN was trained on a training set of 96 samples with SGD with momentum, learning rate 0.01, and the Huber loss. Recall that each sample uses the same map, but with different boundary conditions. We used 32 equally spaced influx locations along the boundaries of the elevation map. The outflux locations were

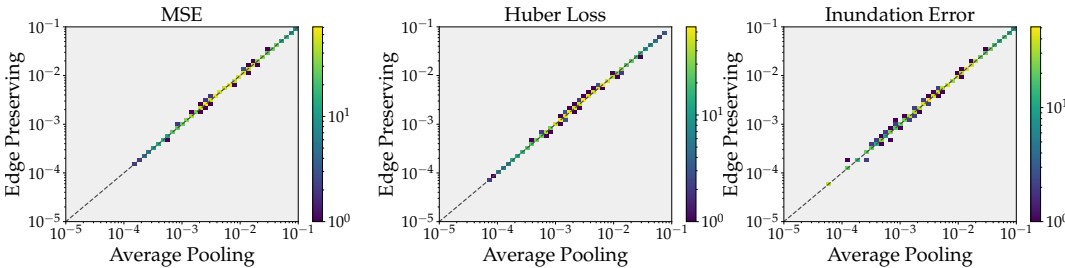

Figure A.2: A comparison of the per-sample loss values of average pooling (x-axis) and edge preserving smoothing followed by average pooling (y-axis). The colorscale is logarithmic. The two methods have a similar per-sample loss values.

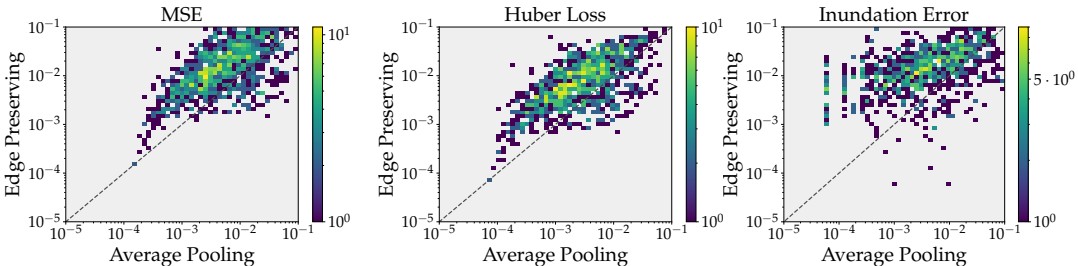

Figure A.3: A comparison of the per-sample loss values of average pooling (x-axis) and edge preserving smoothing followed by max pooling (y-axis). The colorscale is logarithmic. Average pooling typically has a lower loss.

defined in the middle of the edge opposite that of the influx. Each location was experimented with 3 different discharges, for a total of 96 samples. The test set is 100 boundary conditions randomly sampled along the boundaries of the elevation map with a random discharge for each boundary condition. We repeated the experiment in Section 5.2 with different elevation maps randomly selected. Results are shown in Fig. A.5, A.6, A.7, and A.8. A comparison of the per-sample loss values between the baseline and the DNN is provided for each of these examples are shown in Fig. A.4. As can be seen, a DNN trained on a single elevation map with multiple boundary conditions is able generalize well to other boundary conditions. In addition, there are examples where the DNN is not significantly different from the baseline. This can happen when the baseline already works quite well (e.g., as in Fig. A.8).

### A.2.5    Generalization over different elevation maps

We provide more details of the experiment described in Section 5.3. In Table A.1, we provide results which correspond to Fig. 5, along with comparison for different loss functions. This comparison emphasizes that the DNN obtains more significant improvement compared to the baseline for the "hard samples", i.e., the samples with the largest baseline error. This implies that the DNN downsampling is most effective for elevation maps with fine important details that the baseline fails to preserve accurately on a coarse grid.

In addition, we provide a few examples of elevation maps taken from the test set for the DNN trained in Section 5.3. Fig. A.10, A.11, and A.12 shows these examples where we compare the baseline, the DNN, and the DNN correction to the baseline (the ResNet-18 output). As can be seen, the DNN does non-trivial corrections to the map, to preserve hydraulically important details.

### A.2.6    Generalization over long times and geographical regions

In this experiment, we wish to examine the impact of the DNN corrections over long time of hydraulic flow. In addition we examine whether our training method described in Section 5.2 performs well on a different geographical region. We train a DNN over an elevation map of the Ganges river, west of Patna, India, where floods are a major concern every year, during the monsoon season. The elevation map has a resolution of 4 meters × 4 meters per pixel, and contains 2800×2800 pixels, covering an

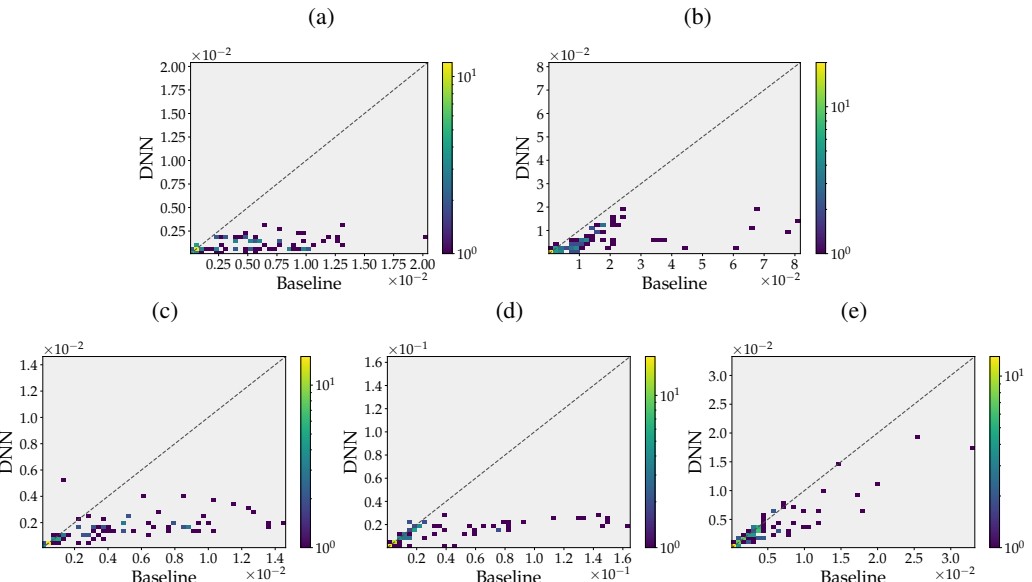

Figure A.4: Huber loss per-sample comparison of the baseline (x-axis) and the DNN (y-axis). Each panel is a DNN trained on a different elevation map. The colorscale is logarithmic. the panels correspond to the following figures – (**a**) Fig. 3 (**b**) Fig. A.5 (**c**) Fig. A.6 (**d**) Fig. A.7 (**e**) Fig. A.8

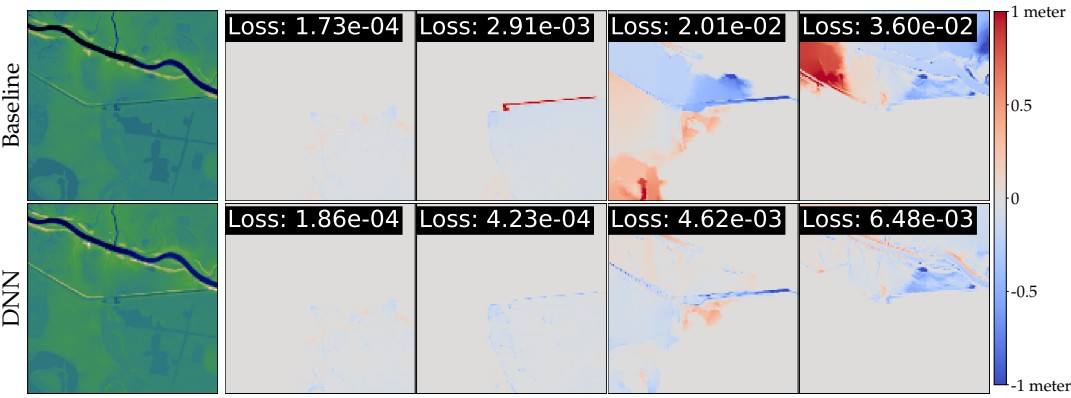

Figure A.5: **Generalization over different boundary conditions.** See Fig. 3 for details.

11.2km×11.2km region. The DNN is trained with the same influx and outflux locations as specified in Sections 5.2 and A.2.4, and with larger discharges that correspond to the much larger Ganges river. Each sample in the training set represents 1 hour of hydraulic simulation. We use the trained DNN to downsample the elevation map into a 175×175 coarse grid map with a resolution of 64 meters × 64 meters per pixel. We then simulate 20 hours of hydraulic flow from two influx locations at the upstream edge of the river and one outflux location at the downstream edge of the river. Results are shown in Fig. A.9. As can be seen, there is a large region on the left side of the map which is inundated only in the baseline solution. In this region, a long, narrow embankment protects a number of villages from flooding; the DNN maintains the correct height of the embankment, but the baseline solution averages out the embankment height with its surroundings, causing substantial overflooding. In addition, a river branch at the bottom right side is (correctly) inundated only in the DNN solution. This area remains dry in the baseline solution because the naive coarsening incorrectly flattens out a narrow channel leading to the river branch. This experiment demonstrates that training over relatively short periods of time is sufficient to significantly improve the accuracy, even when simulating hydraulic flow on a coarse grid for long periods of time. In addition, the training regime of Section A.2.4 seems to generalizes well to a completely different region from which we tuned it on.

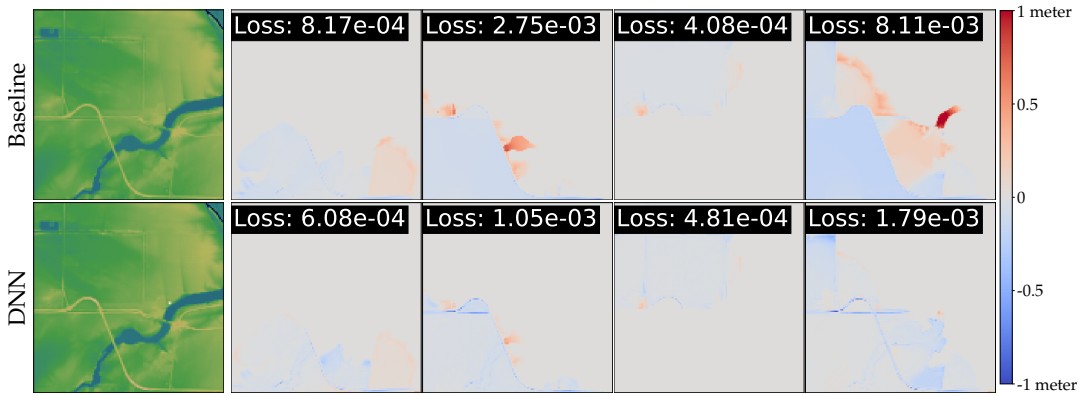

Figure A.6: **Generalization over different boundary conditions.** See Fig. 3 for details.

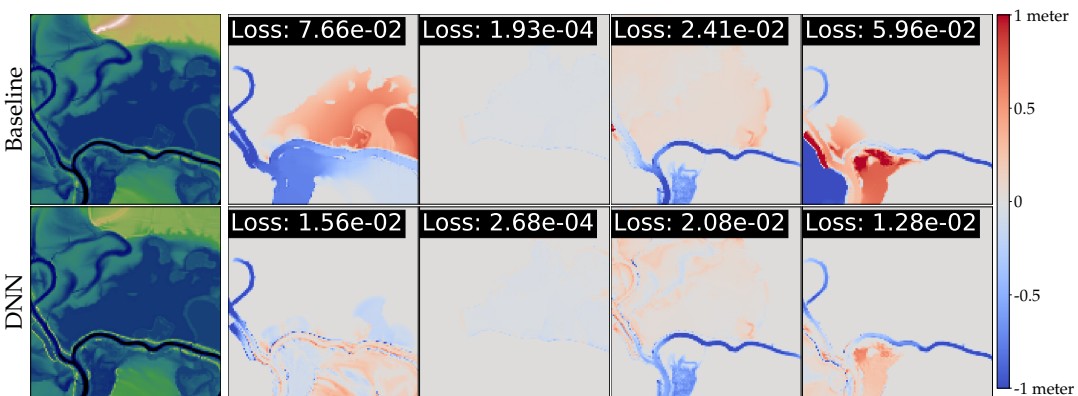

Figure A.7: **Generalization over different boundary conditions.** See Fig. 3 for details.

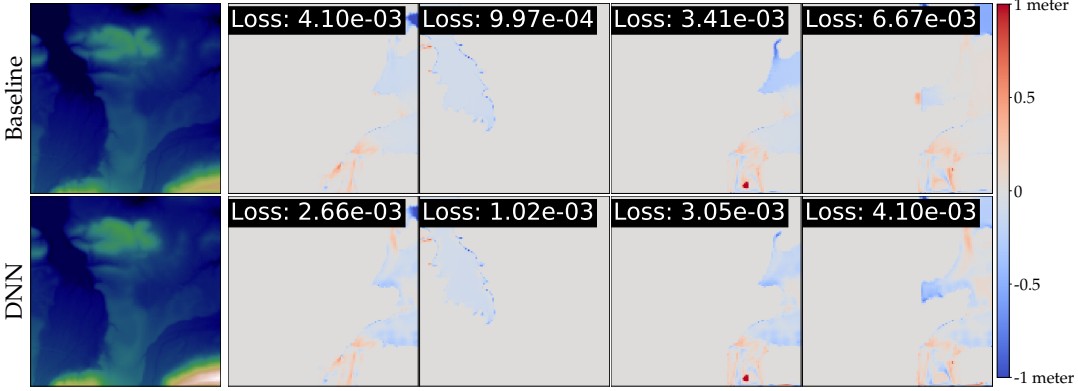

Figure A.8: **Generalization over different boundary conditions.** See Fig. 3 for details.

## A.3 Architecture considerations

We first considered a plain ResNet-18 model with its classifier removed as the downsampling DNN. However, this DNN fails to capture the elevation scale information of the elevation map, meaning that the DNN outputs a coarse grid elevation map with different elevation scale. This failure occurs due to the normalization layers applied in ResNet-18. Normalization layers such as batch normalization [18] or layer normalization [3] are commonly used in modern DNNs, and it has been shown numerous

Table A.1: **Generalization over different elevation maps.** A comparison between average pooling and the DNN trained in Section 5.3 for different loss functions and percentiles. Notably, on samples with a large error, the DNN obtains more significant improvement (compared to the baseline).

| Loss | Average pooling | DNN | Error ratio |
|---|---|---|---|
| Huber loss | $6.04 \cdot 10^{-3}$ | $3.07 \cdot 10^{-3}$ | 1.97 |
| Huber loss 50 percentile | $1.09 \cdot 10^{-2}$ | $5.11 \cdot 10^{-3}$ | 2.13 |
| Huber loss 90 percentile | $2.80 \cdot 10^{-2}$ | $1.10 \cdot 10^{-2}$ | 2.55 |
| MSE | $1.37 \cdot 10^{-2}$ | $6.64 \cdot 10^{-3}$ | 2.06 |
| MSE 50 percentile | $2.44 \cdot 10^{-2}$ | $1.12 \cdot 10^{-2}$ | 2.18 |
| MSE 90 percentile | $6.63 \cdot 10^{-2}$ | $2.69 \cdot 10^{-2}$ | 2.92 |
| Inundation error | $9.77 \cdot 10^{-3}$ | $4.44 \cdot 10^{-3}$ | 2.20 |
| Inundation error 50 percentile | $1.89 \cdot 10^{-2}$ | $8.14 \cdot 10^{-3}$ | 2.32 |
| Inundation error 90 percentile | $5.64 \cdot 10^{-2}$ | $1.77 \cdot 10^{-2}$ | 3.19 |

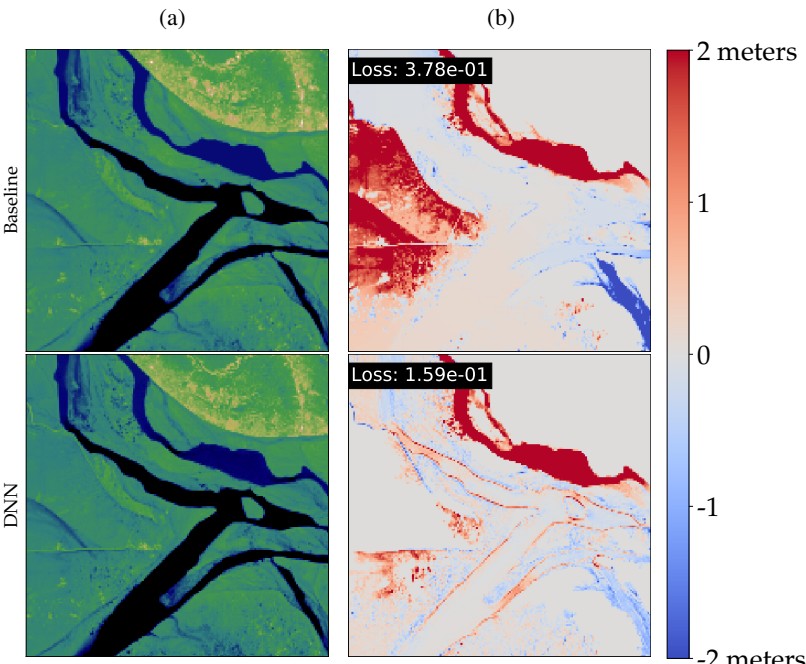

Figure A.9: **20 hours of hydraulic flow over a different geographical region.** A comparison of the solution obtained on a coarse grid between the baseline and a DNN trained on samples of 1 hour of hydraulic flow. Water flows in from bottom left and top left edges of the river, and flows out for the right edge of the river. **(a)** The coarse grid elevation map of the baseline (top) and the DNN (bottom). **(b)** Difference map between the coarse and fine grid solutions along with the Huber loss. Red indicates pixels more inundated in the coarse grid solution, and blue indicates pixels more inundated in the fine grid solution. The solution calculated on the DNN elevation map achieve better accuracy although the DNN was trained on much shorter time periods and a completely different region from which we tuned the training regime.

times that using normalization layers can improve the optimization and generalization of neural networks. However, for the DNN to extract features of the elevation map in different layers, the elevation scale information of the input has to be propagated. Using normalization layers prevents this elevation scale signal propagation.

To resolve this problem, we removed the normalization layers only in the residual connection to enable signal propagation without scale distortion. Indeed, the elevation scale is preserved with this architecture modification, and better convergence and generalization is obtained. Still, the DNN described in the paper, where a skip connection with average pooling is applied from the ResNet-18 input directly to its output, achieves better performance (achieving better initial performance and

faster convergence rate), compared to the modified ResNet-18 architecture. Therefore, we opted to use the architecture from the main paper.

(a)                                (b)                                (c)

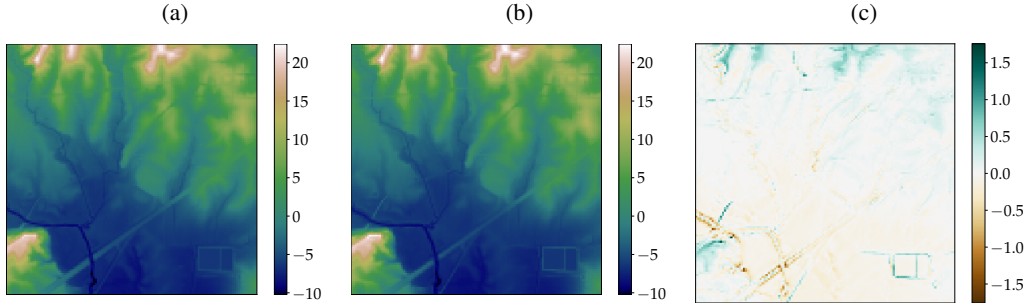

Figure A.10: Example where a narrow water passage (lower left) is expanded. (**a**) – Baseline coarse grid elevation map (**b**) – DNN coarse grid elevation map (**c**) – The correction of the DNN to the baseline. Colorbars in meters.

(a)                                (b)                                (c)

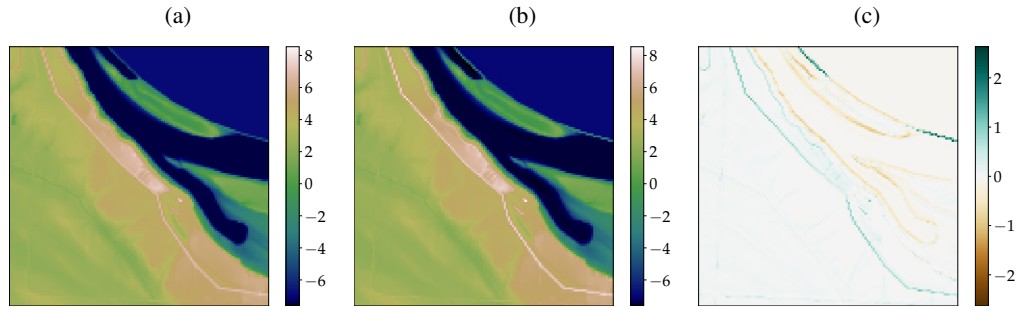

Figure A.11: Example where an embankment height is preserved. (**a**) – Baseline coarse grid elevation map (**b**) – DNN coarse grid elevation map (**c**) – The correction of the DNN to the baseline. Colorbars in meters.

(a)                                (b)                                (c)

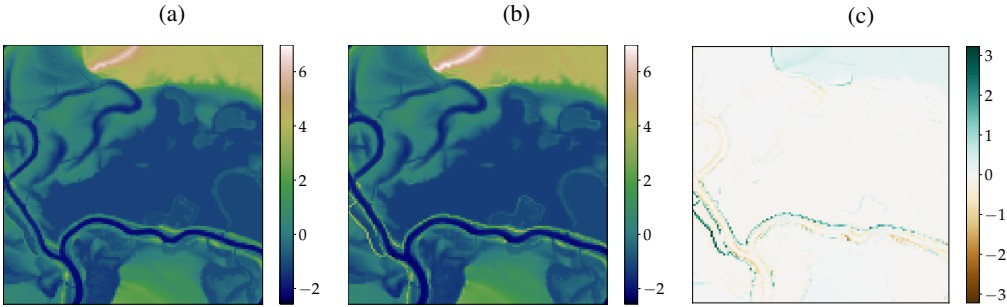

Figure A.12: Example where both river bank is expanded and embankments are preserved next to the river. (**a**) – Baseline coarse grid elevation map (**b**) – DNN coarse grid elevation map (**c**) – The correction of the DNN to the baseline. Colorbars in meters.