# OpenReview forum: "Physics-Aware Downsampling with Deep Learning for Scalable Flood Modeling"
_NeurIPS.cc/2021/Conference — NeurIPS 2021 Poster_

### Official Review · Reviewer_Vnr6 · 2021-07-16

**Rating:** 7
**Confidence:** 5

**Summary:**

This is an interesting paper in which the authors introduce a new downsampling method for generating coarse elevation maps. A new framework was also built to incorporate the governing PDE (and PDE solver) in the training process. The authors have shown the effectiveness of this proposed method on a dataset with simulated water height at fine resolution.

**Limitations And Societal Impact:**

This paper uses deep learning methods for an important problem of flood mapping. I hope to see that this work to be applied to larger regions, e.g., national scale, to aid in flood control. The proposed method can also be potentially used for other applications where downsampling is needed and a governing PDE is given.

**Main Review:**

I have a few comments on this paper.
Q1. Although the use of PDE shows a lot of promise, I am wondering whether this brings any negative impact. For example, PDEs may use simplified or approximate representation and thus observed samples (not simulations) may not strictly follow the PDE. If this is also true for flood prediction, the authors may discuss how this issue may affect the proposed method.
Q2. How accurate it is when the numerical solver is used at the coarse resolution? The PDE represents the continuous relationships that underlie the system. If the samples are too coarse, I am concerned that whether the PDE solver (e.g., using Taylor expansion) is still able to produce good simulations since it has no information about the values in between of two points. I totally agree that downsampled data is helpful in saving the time cost for simulation, but just worried about the accuracy.
Q3. How will other meteorological drivers play a role in flood mapping using this method? It is intuitive that rainfall may be a key factor for the dynamics of the water depth. It would be great to discuss whether the proposed method can also leverage such data sources (e.g., by combining the generated elevation map and other data sources).
Q4. While the authors have already shown the effectiveness of the proposed algorithm, it would be more convincing if more comparisons can be conducted agaisnt existing machine learning approaches (w/ or w/o using PDEs).


**Time Spent Reviewing:**

2

---

> ### Author Response · Authors · 2021-08-10
> **Reply to Reviewer Vnr6**
>
> We thank the reviewer for the positive and helpful feedback to improve the paper.
>
> **Q1**
>
> It is true that PDEs do not capture the full scope of the underlying physics. However, using PDEs is still the most reliable tool for inundation modeling [A, B].
> Observed samples can be combined with the physical model using data assimilation methods for improved forecast. Models that solely count on observed samples might be limited, especially on a large scale, since most observations are derived from satellite data such as NDWI or synthetic aperture radar (SAR). Moreover, there are many challenges in extracting useful data out of these remotely sensed data sources such as depth information.
>
> [A] - Teng, Jin, et al. "Flood inundation modelling: A review of methods, recent advances and uncertainty analysis." Environmental modelling & software 90 (2017): 201-216.
>
> [B] - Di Baldassarre, Giuliano. Floods in a changing climate: inundation modelling. Vol. 3. Cambridge University Press, 2012.
>
> **Q2**
>
> Numerical modeling entails a discretization error that directly depends on the spatial resolution. In addition, input errors such as errors in the elevation map, also depend on the spatial resolution [C]. Indeed, the accuracy deteriorates as the spatial resolution degrades. This paper suggests a method that ideally would reduce the input error. Empirically, we found that the input error is dominant in our setting of flood modeling, but it is an interesting question to quantify this more precisely. We will discuss this question in the main paper.
>
> [C] - Solomon, Justin. Numerical algorithms: methods for computer vision, machine learning, and graphics. CRC press, 2015.
>
> **Q3**
>
> The data used throughout this paper is limited to water flux coming from the boundaries, simulating riverine floods. We agree that more data input such as rain is valuable, and the proposed method is not limited to a single type of boundary conditions. Since the submission of the paper, we improved the code readability and added broader support of boundary conditions such as rain. These changes will be submitted in the next few weeks.
>
> **Q4**
>
> Existing machine learning approaches that use the PDEs, such as [4, 15, 21, 38], modify the PDE solver itself by incorporating a NN into the PDE solver. Importantly, the elevation map has to be represented on a coarse grid regardless of the proposed method. Orthogonally, this paper suggests using a NN to modify the external environment of the PDE. We are not aware of other machine learning approaches that use the PDEs and are easily applicable to flood modeling. Approaches that do not use a PDE are prone to predictions that do not follow physical laws. In addition, such methods are less informative, e.g. lack of depth information.

---

### Official Review · Reviewer_R5nW · 2021-07-16

**Rating:** 7
**Confidence:** 3

**Summary:**

The paper investigates the problem of downsampling high-resolution elevation maps for more computationally feasible flood modeling simulations. For this purpose, a new pipeline is introduced, that features a downsampling network for the elevation maps, and a differentiable PDE solver for the 2D shallow water equation. The authors also provide their simulated large-scale dataset of flood modeling simulations. Finally, the proposed method is evaluated on 1) an example case that highlights the ability to backpropagate gradients through a large number of simulation steps, 2) an experiment with different boundary conditions, and 3) a generalization experiment to new elevation maps.

**Limitations And Societal Impact:**

Throughout the paper and appendix, limitations and aspects that could be improved upon in future work are sufficiently addressed and clarified. One aspect, that could have been investigated are failure cases where the downsampling network overlooks crucial areas in the elevation map, which the average pooling baseline did capture. I assume such cases would only occur in an out-of-distribution test scenario though.

Apart from such failure cases, I do not see any clear potential negative societal impact of the proposed method that should have been discussed.

**Main Review:**

**Originality:** In terms of machine learning, the paper mainly uses existing techniques, however the differentiable implementation of the 2D shallow water equations is interesting. The main novelty is the general approach to the problem. Compared to previous approaches that mainly focus on improving the PDE solver, this paper focuses on improving the input to the solver, which leads to a quite unique perspective. Existing work is properly discussed and even nicely categorized.

**Quality:** The learning setup, loss function, and network architecture are well motivated.
I am not an expert on flood modeling techniques, but I think the experiments are set up thoroughly and produce informative results. Especially, that the network clearly learns to isolate the modified embankment in Section 5.1 over such a large number of simulation and backpropagation steps is impressive. Over the results shown in the main paper and the appendix, the method seems to outperform the intuitive average pooling baseline; not by a huge margin, but quite consistently.

**Clarity:** Overall, the paper is easy to understand, and written and structured well. Only two minor aspects could be described more clearly. First, in Figure 1 a downsampling operation should be added between “True Fluid State” and “Loss Function”, such that the Figure better matches the description in the text. Second, the footnote or the text in Section 4.4 regarding the Huber loss should mention how both map pixel values are combined (this is only clear with the appendix). Otherwise the description in the text mentions *L* as a function of two arguments, while the footnote uses *L* as a function of one argument.

**Significance:** The paper focuses on the rather specific problem of flood modeling, but that in itself already has some direct practical implications and applications as discussed in Section 7. Furthermore, there are some ideas that might be applicable to other PDEs and solvers as well, like the general usage of the differentiable solver within the pipeline, and the perspective of improving inputs to solvers, compared to improving the solver itself.

Overall, the paper provides a novel perspective on flood modeling where deep learning augments a 2D shallow water simulation. In my opinion, this work makes some solid contributions and only has minor weaknesses, leading to my overall evaluation as accept.

------------------------------------------------------------------------------------------------------------------------------------
***After rebuttal:*** After reading the authors response and the other reviews, I still think the paper is interesting and only has minor weaknesses. As a result, I see no reason to update my initial evaluation of accept.

**Time Spent Reviewing:**

4 hours

---

> ### Author Response · Authors · 2021-08-10
> **Reply to Reviewer R5nW**
>
> We thank the reviewer for the positive and helpful feedback to improve the paper.
>
> We accept the suggestions of clarity and will update the paper accordingly to clarify both Figure 1 and the notation of Huber loss.
>
> **Failure cases where the downsampling network overlooks crucial areas**
>
> For the maps we examined, we did not see cases where the downsampling DNN overlooks large areas. However, there are usually small parts of the generated coarse elevation map which are sub-optimal (e.g., when the DNN over-emphasizes embankments to prevent floods in larger regions). This can happen not only in a few samples where the baseline slightly outperformed the modified elevation (as can be seen in Figure 5.b), but also in samples where the DNN was better. Therefore, it is an important question of finding what are the crucial areas which critical for flood predictions (e.g., populated regions). Given a prior knowledge of these locations, it is possible to weight such areas differently during the training process to ensure the DNN does not overlook them.

---

### Official Review · Reviewer_EGwq · 2021-07-17

**Rating:** 6
**Confidence:** 3

**Summary:**

The paper presents a deep learning-based method for coarse-grain discretization in solving the PDEs of flood modeling. A ResNet is used to downsample the input terrain, and the coarse-grain output is then fed into a numerical solver. A fine-grain numerical solver generates the ground truth for end-to-end training of the downsampling neural network. The work also builds a dataset of elevation maps and their flood simulation results.  The experiment shows that the proposed method surpasses the baseline that uses the average pooling downsampling.

**Limitations And Societal Impact:**

Seems adequate.

**Main Review:**

Pros:
- The idea of integrating discretization and numerical solver in an end-to-end trainable system seems interesting.

- The paper is mostly well-written and easy to follow.

Cons:
- The work lacks clear motivation for developing such a solver-aware downsample learning strategy. As mentioned in the related work, there have been several learning-based coarse-grain discretization methods in the prior work. It is unclear why the previous discretization methods are insufficient for flood modeling.

- While learning solver-aware downsampling seems to be a sensible design, the paper lacks details on some of its main components: 1) What is the coarse grid representation $\hat{z}$? and what are the different coarse grids formed by the network output? 2) What is the computation graph of the numerical solver, and how the gradient is back-propagated? 3) The simulation iterations are inconsistent in different places: in Line 225, it says one hour of hydraulic flow, while in Line 283, it uses 6 hours. What about using 1-hour short-run in Sec 5.1?

- The main paper did not include the run-time complexity of training and inference for the overall system.

- The experimental evaluation is lacking in multiple aspects and the overall claims are not very convincing. First of all, the method is compared with a simple baseline (average pooling) only and misses other more related work such as [4] and [A]. Secondly, it is unclear why the flow accuracy uses the Huber loss, which seems to downplay the large errors. Finally, Figure 5 did not provide detailed quantitative results and those results can only be found in the appendix. For the target application, it is unclear how significant the errors are for the flood modeling.

[A] Mishra, Siddhartha. “A Machine Learning Framework for Data-Driven Acceleration of Computations of Differential Equations.” Mathematics in Engineering 1.1 (2018): 118–146.

Post rebuttal: The author's rebuttal addressed most of my concerns and I would like to raise my rating to the positive side.

**Time Spent Reviewing:**

2

---

> ### Author Response · Authors · 2021-08-10
> **Reply to Reviewer EGwq**
>
> We thank the reviewer for the helpful feedback. Below, we address the concerns the reviewer raised.
>
> **No clear motivation. Why won't the previous works do the job?**
>
> This is explained in the related works section, lines 103-114. The key point is that previous works [A,4,15,21,38] focused on improving the PDE solver, while this paper focuses on improving the input to the solver. Specifically, previous works aimed to modify the PDE solver, to enable using a coarser grid (potentially with some overhead) while retaining accuracy and numerical stability. However, they did not examine how to optimally coarse grain the external environment (given as input to the solver)—as we do here, for terrain elevation maps in the context of flood modeling. In this case, it is critical that the coarse-grained map retains the relevant features, to prevent drastic differences between fine and coarse solutions (e.g., floods). Moreover, the two approaches (modifying the PDE solver vs. modifying the external environment) are orthogonal, and can be potentially combined in future work.
>
> **What is the coarse grid representation Z? What are the different coarse grids formed by the network output?**
>
> The notation is described in section 4.1. Z is the terrain elevation map given at a fine grid resolution, while \hat{Z} is the downsampled (i.e. coarse grid) version of it, produced by the deep network output. Both the input and output maps have ordinary square grids, with the output is a coarser version of the input (x16 coarser in each dimension).
>
> **What is the computation graph of the numerical solver? How the gradients are back-propagated?**
>
> The numerical solver is a finite difference discretization of the shallow water equations, as described in section 3, lines 132-135 - “We follow De Almeida & Bates [7] to numerically approximate eqs. 1 and 2 from the continuous domain into the discrete domain, where the numerical solution uses a finite difference scheme applied to a staggered grid”. The gradients are back-propagated through automatic differentiation using PyTorch. Complete technical details can be found in De Almeida & Bates [7] and our published code, which can be found at the bottom of page 2.
>
> **The simulation iterations are inconsistent.**
>
> As described in section 4.4 (lines 225-226), we always trained the model on 1 hour of hydraulic flow. However, Section 5.1 does not consider training. There we simulate 6 hours of hydraulic flow to demonstrate that gradients can propagate through PDEs for many iterations. This is described in lines 266-267: “Specifically, we show that the gradient signal can properly flow even through tens of thousands of unrolled recurrent units representing the PDE time-steps”.
>
> **Paper did not include run-time complexity of training and inference for the overall system.**
>
> The inference run-time is negligible compared to the solver execution since the elevation map is downsampled by the DNN once, and the solver remains the same.
> The training run-time of the experiments described in Section 5.3 is 6-7 days on a 8-GPU machine. The training run-time of the experiments described in Section 5.2 is 4-5 hours on a 4-GPU machine (technical details can be found in lines 531-534). We note that the utilization of the GPUs during training is much lower than in standard classification tasks, because of sub-optimal implementation of the PDE solver. In addition, since the input elevation maps are large, each GPU can use only up to 4 samples at once (leading to a total batch size of 32 samples, as described in lines 218-219). It is interesting to examine how to optimize the training process to include PDEs more efficiently. The complexity between coarse and fine grids is discussed in lines 39-42.
>
> **The method is compared with a simple baseline and misses other related work such as [4] and [A].**
>
> As we mentioned above, the related work does not tackle the coarsening mechanism of the external environment. [4] suggest a method of discretizing the PDE itself, rather than the elevation map. [A] suggests a similar method to [4], where the input to the NN is already downsampled to a coarse grid, and the question at hand is choosing the numerical scheme coefficients, rather than mapping the external environment.
>
> **Why huber loss?**
>
> We experimented with different loss functions, and provided results and comparison in appendix A.2.2, lines 542-559. As mentioned in the paper - “the correlation of the inundation error with the Huber loss is somewhat higher than the MSE. Also, the Huber loss exhibited better convergence properties”. In addition, the paper provides comparison for the MSE loss in Table A.1.
>
> **Figure 5 didn’t provide quantitative results (only in appendix). It is unclear how significant the errors are for flood modeling.**
>
> Indeed, for Figure 5, additional quantitative results were provided in appendix section A.2.5 and Table A.1. We will update the main paper to include a summary of these results near the figure.

---

> > ### Comment · Reviewer_EGwq · 2021-09-01
> > **Post rebuttal**
> >
> > I thank the author's detailed reply to my initial comments. Most of the original concerns are well addressed by the rebuttal and as a result, I would like to raise my rating. That said, I still feel the connection between the proposed method and the flood application is a bit loose: if the paper presents a general method, it would be more convincing to test it on several problems.

---

### Official Review · Reviewer_48GU · 2021-07-22

**Rating:** 7
**Confidence:** 4

**Summary:**


The paper proposes a method for physics-informed downsampling of the terrain map for scalable inundation modelling.


Main paper contributions are stated to be as following:

- new method for modelling physical processes with deep learning. Specifically, authors combine a deep neural network (DNN) with the partial differential equations (PDEs) describing the flow of water. The DNN downsamples the elevation map, and the downsampled elevation map is fed into the PDEs, where each time step is a recurrent unit, to calculate the water height solution)
-creating of a dataset for a particular problem


**Main Review:**

=== Summary ====

The paper proposes a method for physics-informed downsampling of the terrain map for scalable inundation modelling.


Main paper contributions are stated to be as following:

- new method for modelling physical processes with deep learning. Specifically, authors combine a deep neural network (DNN) with the partial differential equations (PDEs) describing the flow of water. The DNN downsamples the elevation map, and the downsampled elevation map is fed into the PDEs, where each time step is a recurrent unit, to calculate the water height solution)
-creating of a dataset for a particular problem

== Comparison to previous work ==

Applications of neural network models for solving PDEs is not particularly new (Optimally weighted loss functions for solving PDEs with Neural Networks (2020) R. van der Meer, C. Oosterlee, A. Borovykh).  At the same time, this idea was also used for flood prediction, however in quite different setting (https://arxiv.org/abs/1908.10312). Therefore, current approach, with this formulation and application has not been introduced before.

==Theory==

Authors discuss downsampling of elevation maps for efficient shallow water flow modelling. This approach is in the intersection of three important field of research: (1) using neural networks for efficient PDE modelling; (2) using deep learning for efficient downsampling of elevation maps; (3) flood modelling. This problem formulation is equally novel and important.

==Method==

Authors provide detailed explanation of the proposed approach in supplementary material, but I have not seen the code (apologies if it is attached but I have not noticed it).
The proposed model architecture consists of several models, which together estimate the fluid state. Authors use modified ResNet-18 architecture  as downsampling neural network.
In the experiments, authors use the proposed neural network modl to solve the shallow water equations on the coarse grid, over variety of elevation maps and boundary conditions. They show that the proposed method is able to provide accurate flood modelling.

=== Pros ===

The proposed method is novel and interesting, authors provide clear description of the method, experiments and outcome, together with the data and code.
They also provide assessment of broader implications and theoretical contribution on the domain of interest.

===Cons===

The results of current model and comparison to previous work are somewhat scattered in the text. What value this work adds to the traditional methods?
The model, used in this work, is ResNet, which is quite a ‘classical’ deep learning model, therefore the work does not introduce a particularly novel ML method.

=== Basis for recommendation ===

This paper introduces some important concepts and methods and can be published in NeurIPS 2021. However, some clarification is needed to make this work easier to read.

=== Questions to address in the revision ===

Please state explicitly where the code/data can be found
I would be interested to see more detailed explanation of the contribution of how neural networks are combined with PDEs and ow it compares with previous work (add more detailed explanation to lines 90-114) . How current results can be compared to previous work?
Please state explicitly the results, comparison to previous work and to the baseline (these are somewhat scattered in the text). What value this work adds to the traditional methods? Can you please quantify “significant error reduction” (line 320)?


**Time Spent Reviewing:**

2 days

---

> ### Author Response · Authors · 2021-08-10
> **Reply to Reviewer 48GU**
>
> We thank the reviewer for the positive and helpful feedback to improve the paper.
>
> **Code and data availability**
>
> The code and data are available in the link provided in a footnote at the bottom of page 2.
>
> **Contribution and comparison to previous results**
>
> This paper can be compared to previous works [4, 15, 21, 38] in the design of the learning process and the combination of NN and PDEs. These works modify the PDE solver itself by incorporating a NN into the PDE solver. Importantly, the elevation map has to be represented on a coarse grid regardless of the proposed method. Orthogonally, this paper suggests using a NN to modify the external environment of the PDE. The PDE solver remains intact, so there is no computational overhead.
>
> **Can you please quantify “significant error reduction” (line 320)?**
>
> A detailed explanation can be found in Appendix Table A.1, as mentioned in line 320. The main paper will be updated with the following clarification:
> “The modified elevation maps exhibited an improvement of the Huber loss by a factor of 2 over all the tested elevation maps. This improvement increases to a factor of 2.5 as we look only at harder examples (90% percentile). Other accuracy metrics show similar results with a larger improvement factor of up to 3.”

---

### Decision · Program_Chairs · 2021-09-27

**Decision:**

Accept (Poster)

**Comment:**

The paper proposes a method for physics-informed downsampling of the terrain map for scalable flood modeling. The application problem is extremely important and challenging. The proposed solution, while not totally novel, is reasonable. The experiment would be more convincing if thorough evaluation can be conducted with strong baselines in the fields. We hope that the authors can seriously consider improving the evaluation in the final version of the paper.